# Many-body interference in kagome crystals

Chunyu Guo (Mark)[1]✉, Kaize Wang[1], Ling Zhang[1], Carsten Putzke[1], Dong Chen[2,3], Maarten R. van Delft[4,5], Steffen Wiedmann[4,5], Fedor F. Balakirev[6], Ross D. McDonald[6], Martin Gutierrez-Amigo[7], Manex Alkorta[8,9], Ion Errea[8,9,10], Maia G. Vergniory[10,11], Takashi Oka[12], Roderich Moessner[13], Mark H. Fischer[14], Titus Neupert[14], Claudia Felser[3] & Philip J. W. Moll[1]✉

When electrons in metals act collectively, they enable emergent phenomena and electronic functionalities that transcend the behaviour of individual particles[1]. Coherent collective charge motion has so far been observed primarily in superconductors, in which it arises with the formation of Cooper pairs[2,3]. Here we report experimental evidence for coherent charge transport in the normal state of the kagome metal $CsV_3Sb_5$, indicative of a distinct collective electronic state. The signature is a set of magnetoresistance oscillations in mesoscopic crystalline pillars under in-plane magnetic fields, with a periodicity determined by the number of magnetic flux quanta $h/e$ threading between adjacent kagome layers—effectively forming an interlayer Aharonov–Bohm interferometer. The cooperative nature of this phenomenon is evidenced by a non-analytic angular dependence characterized by abrupt transitions between discrete oscillation frequencies and its persistence over length scales that exceed the single-particle mean free path. Notably, the oscillation amplitude matches other anomalous electronic responses reported in $CsV_3Sb_5$, pointing to an underlying mechanism that establishes intrinsic coherence. These findings shed new light on the debated nature of correlated order in kagome metals and establish $CsV_3Sb_5$ as a platform for realizing long-range coherent charge transport in the absence of superconductivity—opening new directions for coherence in correlated electron systems beyond conventional models.

A central goal of contemporary condensed matter physics is the rational design of interacting electronic phases and emergent cooperative quantum phenomena driven by electronic correlations[1]. Achieving fine control over electron behaviour beyond conventional band-structure engineering holds promise for unlocking exotic response functions and enabling future technologies, particularly those making use of quantum coherence. One strategy to amplify the effects of electronic interactions is to quench the kinetic energy of electrons, thus effectively localizing them by engineering quantum materials in which several electronic pathways interfere destructively[4]. This principle underlies the physics of 'flat-band systems', exemplified by magic-angle bilayer graphene[5]. Attention has further turned to other lattice geometries that support frustrated quantum paths, such as the Lieb[6] and kagome lattices[7].

The layered kagome superconductor $CsV_3Sb_5$ ($T_c \approx 2.8$ K) has gained notable attention owing to its intertwined electronic states. Its charge order at $T_{CDW} \approx 94$ K (refs. 8–11) anticorrelates with $T_c$ and, although some studies suggest that its superconductivity is unconventional[12–17], it remains a matter of debate. Perhaps the most compelling open question lies in the enigmatic change of the electronic ground state at an intermediate temperature scale, $T' \approx 30$ K, identified by several experimental probes[7,18–25]. The microscopic nature of this low-temperature state and its broken symmetries (if any) remains unresolved. A prominent hypothesis proposes the emergence of persistent orbital currents, often described as loop currents circulating within the kagome plaquettes[26–30]. Discrete and metastable switchability driven by out-of-plane fields acting on an effectively chiral electronic system have been reported on the basis of tunnelling microscopy[31,32], substantiating the presence of an electronic state with history dependence at low temperatures.

Here we present experimental evidence that the state emerging below $T'$ hosts long-range electronic coherence and directly contributes to coherent out-of-plane charge transport. It manifests through interference effects that persist over macroscopic distances spanning several micrometres and remain robust up to $T'$. The observed interference patterns in magnetoresistance measurements establish that this state actively participates in charge transport. Moreover, the in-plane angular dependence of the oscillation frequencies exhibits a notable switching behaviour, underscoring a non-trivial origin that is difficult to reconcile with a purely single-particle framework. Together, these findings point to a macroscopically coherent collective electronic state reminiscent of superconductivity in its coherence properties, yet absent a dissipationless condensate.

[1]Max Planck Institute for the Structure and Dynamics of Matter, Hamburg, Germany. [2]College of Physics, Qingdao University, Qingdao, China. [3]Max Planck Institute for Chemical Physics of Solids, Dresden, Germany. [4]High Field Magnet Laboratory (HFML - EMFL), Radboud University, Nijmegen, The Netherlands. [5]Institute for Molecules and Materials, Radboud University, Nijmegen, The Netherlands. [6]National High Magnetic Field Laboratory, Los Alamos National Laboratory, Los Alamos, NM, USA. [7]Department of Applied Physics, Aalto University School of Science, Aalto, Finland. [8]Centro de Física de Materiales (CFM-MPC), CSIC-UPV/EHU, Donostia-San Sebastian, Spain. [9]Fisika Aplikatua Saila, Gipuzkoako Ingeniaritza Eskola, University of the Basque Country (UPV/EHU), Donostia-San Sebastian, Spain. [10]Donostia International Physics Center, Donostia-San Sebastian, Spain. [11]Département de Physique et Institut Quantique, Université de Sherbrooke, Sherbrooke, Quebec, Canada. [12]The Institute for Solid State Physics, The University of Tokyo, Kashiwa, Japan. [13]Max Planck Institute for the Physics of Complex Systems, Dresden, Germany. [14]Department of Physics, University of Zürich, Zürich, Switzerland. ✉e-mail: chunyu.guo@mpsd.mpg.de; philip.moll@mpsd.mpg.de

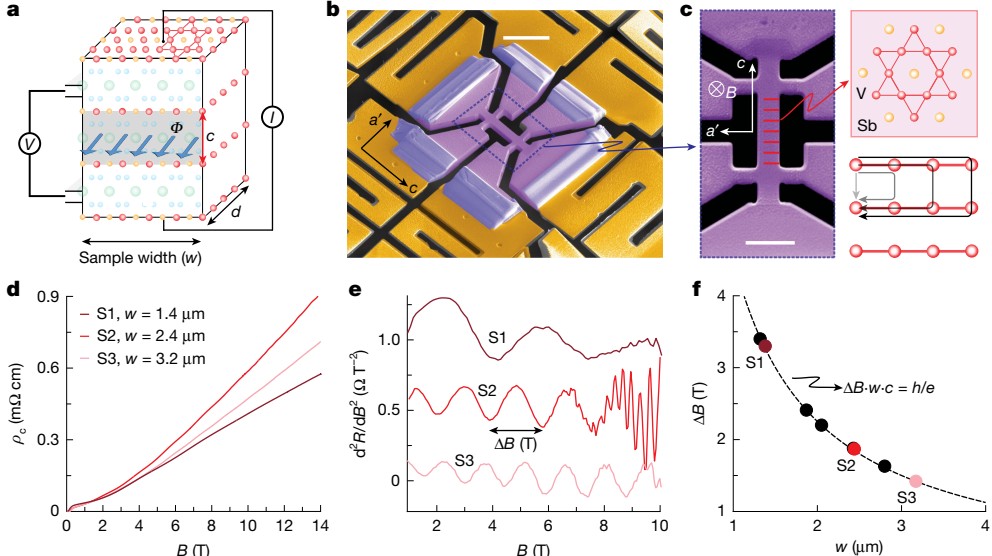

**Fig. 1 | $h/e$ oscillations in CsV$_3$Sb$_5$. a**, Sketch of the transport bar for out-of-plane transport. The in-plane field threads flux $\Phi$ between adjacent kagome planes. **b**, Scanning electron image of the CsV$_3$Sb$_5$ microstructure supported by soft membrane springs. **c**–**e**, Close-up of the device (**c**), indicating the kagome layers (red), magnetoresistance (**d**) and its second derivative of devices S1, S2 and S3 of varying width (**e**). **f**, The oscillation periods of samples S1–S3 as well as further samples outlined in the Supplementary Information. The period is inversely proportional to the device width and the proportionality is given by the flux quantum alone (dashed line is fitting-parameter-free). Scale bars, 10 μm (**b**); 5 μm (**c**).

## Observation of $h/e$ oscillations in CsV$_3$Sb$_5$

The central observation of this study is the emergence of field-periodic oscillations in the interlayer magnetoconductance of micron-sized pillars (Fig. 1), which effectively act as macroscopic stacks of kagome planes. An electric current is applied along the pillar axis, perpendicular to the kagome layers, whereas a magnetic field $B$ is oriented in plane perpendicular to the sample surface. At 2 K and above, the superconducting upper critical field $H_{c2}$, the magnetoresistance exhibits a superlinear increase, superimposed with pronounced oscillations that are periodic in the magnetic field (Fig. 1e). These oscillations persist at higher fields but gradually merge into $1/B$-periodic Shubnikov–de Haas oscillations, which eventually obscure their distinct identification in the high-field regime.

The oscillation period $\Delta B$ is uniquely determined by the magnetic field required to thread a single flux quantum between adjacent kagome planes in the rectangular stack, following the relation $\Delta B \cdot w \cdot c = h/e := \Phi_0$. This combines the atomic-scale interlayer spacing ($c \approx 9$ Å) with the macroscopic device width $w$, which is precisely controlled during fabrication and typically ranges from 1 to 3 μm. Unlike Shubnikov–de Haas oscillations that encode the fermiology and chemical potential of a material, the $h/e$ period is strongly independent of material details besides the interlayer spacing, suggestive of a more general quantum process at its origin. Varying the width systematically between devices allows us to unambiguously identify this universal scaling relation (Fig. 1f; note that the dashed line is fitting-parameter-free). Without any model considerations, this observation directly evidences a sensitivity of the electronic system to in-plane confinement over macroscopic distances greater than 3 μm.

This observation in CsV$_3$Sb$_5$ relies on two recent methodological advances. First, detecting a measurable signal requires averaging over a macroscopic number of identical kagome layers. To achieve this, we use focused ion beam (FIB) machining to sculpt bars aligned along the $c$ direction from the plate-like single crystals. Second, CsV$_3$Sb$_5$ is known to be exceptionally sensitive to strain[18,20,33], necessitating careful mechanical decoupling to avoid unintentional strain from thermal mismatch with the substrate[34]. This is accomplished by suspending the devices on soft SiN$_x$ membranes coated with gold for electrical contact, as demonstrated previously[18,20,33]. Indeed, the oscillations are suppressed in substrate-supported devices under typical strain levels comparable with those in experiments on bulk crystals (see Methods).

Notably, the relevant magnetic flux is set by the spacing between adjacent kagome planes, rather than by the crystallographic unit cell reconstructed by charge order. Below $T_{CDW}$, the interlayer periodicity undergoes a transition to a double-layer ($2 \times 2 \times 2$) or quadruple-layer ($2 \times 2 \times 4$) superstructure[35,36]. The precise nature of this superstructure remains debated, as calculations predict nearly degenerate energies for both configurations, consistent with experimental reports of coexisting states and first-order transitions between them[11,35,37]. Despite the clearly established presence of an out-of-plane reconstruction, our measurements uniquely identify the nearest-neighbour interlayer spacing as the relevant length scale—again underscoring the insensitivity of the observed oscillations to band-structure details associated with the charge-ordered state.

## Exotic angular dependence

So far, magnetic fields have been applied perpendicular to the sample surface. We now discuss a marked discontinuity in the oscillation behaviour when the magnetic field is rotated within the kagome plane. At fixed field magnitude, the direction of the field is tilted away from the surface normal ($\varphi = 0°$) within the kagome plane (Fig. 2). At low angles, the quantum interference process continues to count flux quanta between the planes and the measured periods are found to grow accordingly as $\Phi(\varphi) = \Phi_0 \cos(\varphi)$. Notably, this smooth trend breaks down abruptly at $\varphi = 45°$, at which the oscillation period exhibits a discontinuous jump. This transition is clearly visible in the raw data even without Fourier analysis (see Fig. 2b insets, spaced by 10° increments). As the field is further tilted towards the orthogonal sidewalls ($\varphi = 90°$), the oscillation period once again follows a $\cos(\varphi)^{-1}$ scaling but now with an effective width corresponding to the shorter sidewall (green branch in Fig. 2).

The discontinuous switching strongly contrasts the smooth flux-tuning characteristic of interferometric processes. When several independent processes coexist, such as one per sidewall, several frequencies are typically observed simultaneously. This is common in single-particle quantum phenomena such as de Haas–van Alphen oscillations, in which

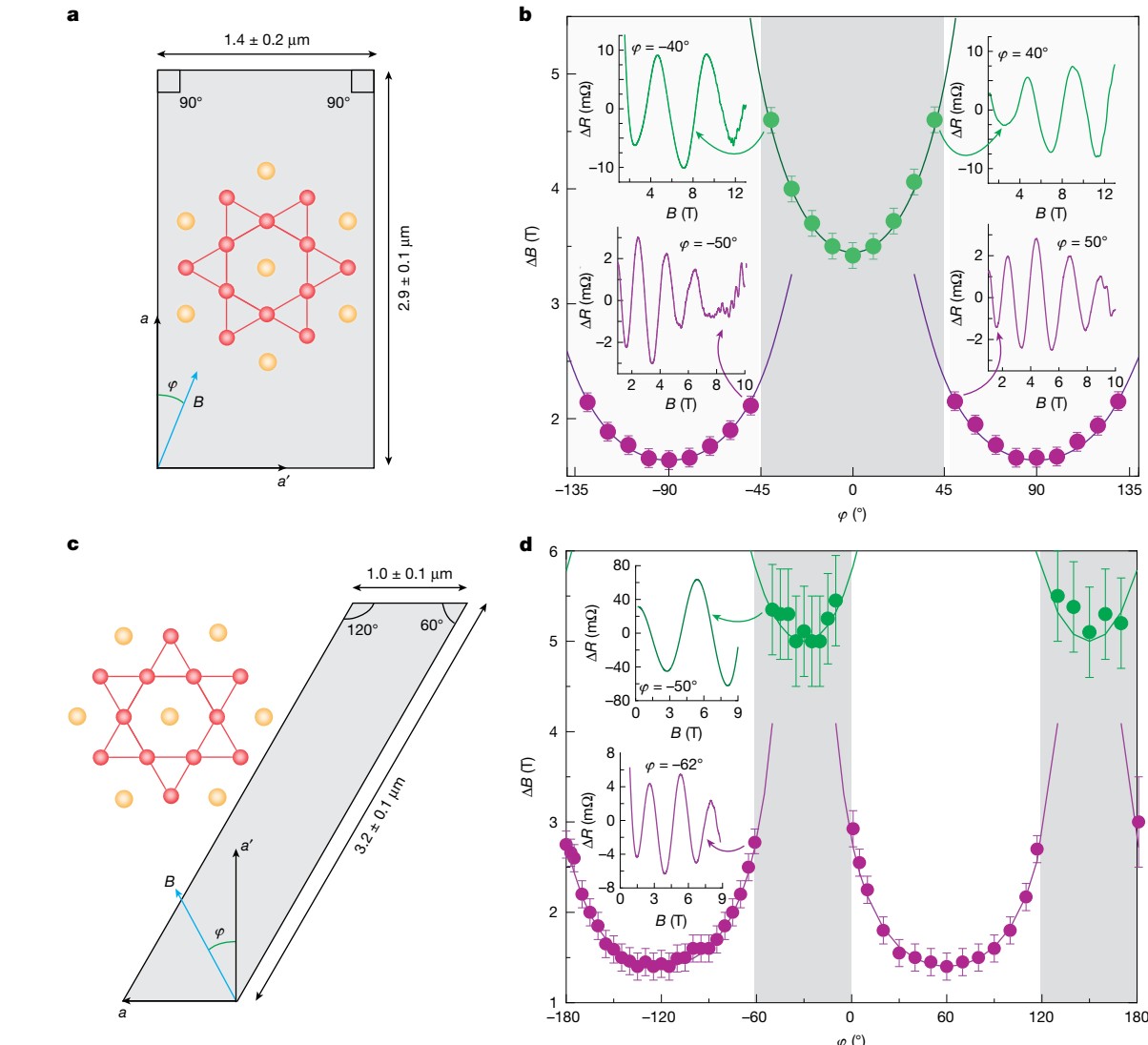

**Fig. 2 | Angular dependence: in-plane rotation. a**, Illustration of field rotation within the kagome plane. $\varphi$ is defined as the angle between the magnetic field and the crystalline *a* direction. **b**, Angular dependence of oscillation period in device S1. Two distinct oscillation regimes appear, corresponding to the two surface dimensions (green, 1.4 μm; purple, 2.9 μm). The oscillation period jumps abruptly at both −45° and 45°. Insets, raw data after background subtraction for ±40° and ±50°, showing a clear jump in frequency. **c,d**, Same as **a** and **b** but for the parallelogram-shaped sample. The two oscillation regimes persist but are clearly modified by the geometry.

distinct frequencies from different regions of the Fermi surface produce characteristic beating patterns. By contrast, a single frequency is observed at all angles with perfect reproducibility in all studied samples (see Supplementary Information). At the switching point, interference from one surface abruptly vanishes as the other emerges. This behaviour demands coherent communication and non-local coupling between surfaces, implying that one process actively suppresses the other. Such coordination necessitates a long-range coherent many-body state.

This leaves the important question of what sets the switching point. The Fermi surface alone can be ruled out, as the switching angles in rectangular samples occur in 90° intervals, which is incompatible with the symmetry of the kagome lattice. To test how the geometry influences the switching angles, a sample with a parallelogram-shaped cross-section was fabricated, designed with interior angles of 60° and 120° to match the symmetry of the kagome crystal structure and eliminate effects related to surface termination (Fig. 2c). A notable result emerged: the large oscillations again follow the same $\cos(\varphi)^{-1}$ scaling, with the period variation set by the different widths of both sidewalls (Fig. 2d). This poses important constraints on the microscopics of the

many-body state. First, a rectangular boundary is not essential for interferometry. Second, the shape is a key factor for the switching angles, as—in both cases—the switching angles match the internal angles of the geometry (rectangle: 90°; parallelogram: 60°, 120°). Third, even when the shape modifies the angle spectrum substantially, oscillations do not coexist and discrete switching is a robust feature of this material. The absolute location of the switching angles provides further insights. Switching occurs at the surface normal of the short segments ($\varphi = 0°$), in contrast to the rectangle, which featured $\cos(\varphi)^{-1}$ dispersions centred around the surface normal. This demonstrates that not a single factor dictates the switching but, rather, several factors, including the surface termination, the sample shape and the crystal orientation, conspire to create a non-trivial energy landscape for the many-body state.

## Beyond single-particle ballistic transport

To understand the observed periodic-in-*B* oscillations, a first critical step is to narrow the physical regime in which microscopic theories might describe such oscillations. At first glance, a natural comparison

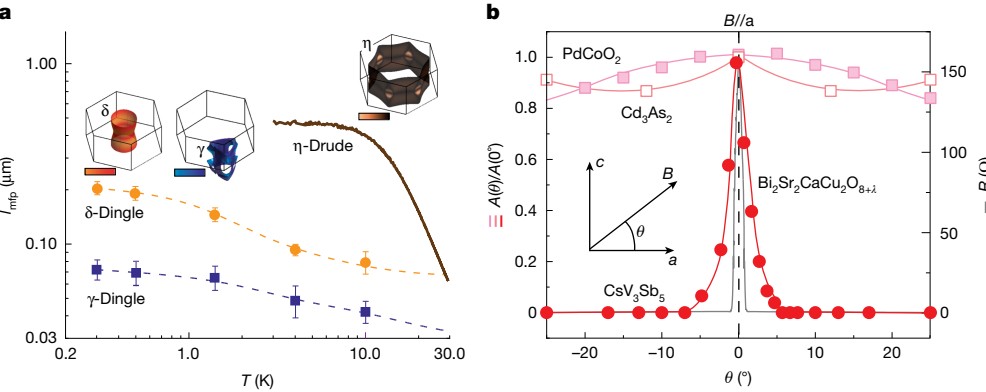

**Fig. 3 | Mean free path analysis and angular dependence of oscillation amplitude. a**, Quantum and transport mean free paths extracted from a Drude and a Dingle analysis, respectively. The insets present different branches of Fermi surfaces in the Brillouin zone and the colour bar stands for the distribution of Fermi velocity anisotropy ($v_F^{ip}/v_F^z$). **b**, Angular dependence of $h/e$ oscillation amplitude (circles), rotating the field out of the kagome planes. This is in strong contrast to the slow and smooth evolution of $h/e$ oscillations in PdCoO$_2$ (ref. 38) and Cd$_3$As$_2$ nanowire[43], whereas correlated phenomena such as vortex flux oscillations in layered superconductors show sharp features for in-plane alignment[44].

arises with $h/e$ flux-periodic oscillations reported in the same experimental geometry for the ultra-clean layered metal PdCoO$_2$ (ref. 38). In that context, single-particle models have been proposed on both semiclassical ballistic[39] and fully quantum mechanical descriptions based on interlayer interference. Although PdCoO$_2$ and CsV$_3$Sb$_5$ share the same $h/e$-per-layer periodicity, this alone does not imply a common underlying mechanism. Fundamentally, any cyclic quantum process in a magnetic field is governed by the flux quantum, as seen in diverse phenomena including Aharonov–Bohm oscillations, Hofstadter patterns and de Haas–van Alphen oscillations. Although the behaviour in PdCoO$_2$ is probably well captured by single-particle physics, we argue that the observations in CsV$_3$Sb$_5$ instead require invoking the coherence of a collective many-body state.

This conclusion is based on important qualitative differences between the observations in PdCoO$_2$ and CsV$_3$Sb$_5$ that demonstrate such fundamentally distinct physics. First, PdCoO$_2$ exhibits several oscillation frequencies at all field angles, a natural outcome of superposed single-particle processes contributing collectively to the transport signal. Accordingly, the $\cos(\varphi)^{-1}$ dependence of several independent processes in PdCoO$_2$ evolves smoothly across the entire angular range, consistent with a process governed only by the effective magnetic flux. By contrast, CsV$_3$Sb$_5$ shows a discrete switching between oscillation frequencies, which fundamentally cannot be captured by any existing theories developed for PdCoO$_2$.

A second key distinction lies in the electronic mean free path. PdCoO$_2$ is one of the most conductive oxides known and exhibits unusually long transport mean free path exceeding 20 μm at low temperatures, naturally hosting single-particle trajectories traversing the full sample width unimpeded. This condition is essential for any model based on single-particle interference, whether quantum or semiclassical, to account for sample-size-dependent oscillations. Indeed, previous studies on PdCoO$_2$ confirmed that the oscillations vanish when the transport mean free path is reduced below the sample dimensions through electron irradiation[38]. By notable contrast, CsV$_3$Sb$_5$ exhibits oscillations that persist over length scales far exceeding both its quantum and transport mean free path, as we will detail in the following.

We performed systematic in-plane transport measurements on a series of bars with varying widths. From these measurements, we extracted a transport mean free path of approximately 560 nm at 2 K (see Supplementary Information for the complete transport mean free path discussion). Given the multiband nature of CsV$_3$Sb$_5$, band-dependent scattering times could, in principle, affect such estimates. To account for such effects, we applied several distinct methods to estimate the transport mean free path on both microstructures and bulk crystals. All approaches consistently yielded transport mean free paths on the order of 500 nm. This value is in line with expectations, considering the known levels of chemical disorder and the presence of domain boundaries in the charge-ordered state of CsV$_3$Sb$_5$, and it further agrees well with other experimental reports[40]. Crucially, no experimental evidence supports a transport mean free path as large as 3 μm, the length scale over which oscillations are still robustly observed. More experiments on microstructures intentionally fabricated from lower-quality crystals confirmed the persistence of $h/e$ oscillations, even when the sample dimension exceeds the transport mean free path by a factor of ten (see Supplementary Information). These findings clearly rule out a semiclassical ballistic scenario in which wavepackets traverse the sample coherently without scattering.

This leaves quantum-coherent mechanisms as an alternative, in which the magnetic field enters not through the Lorentz force but by means of a Peierls substitution of the vector potential, leading to genuinely interferometric effects at the origin of the oscillations[38]. It is well established that elastic scattering does not lead to decoherence, as exemplified by the Aharonov–Bohm effect, in which electronic wave coherence is preserved despite frequent elastic boundary scattering along a ring structure[41]. Thus, the phase coherence length may well exceed the transport mean free path, but this does not hold for CsV$_3$Sb$_5$. The presence of Shubnikov–de Haas oscillations enables the extraction of the quantum mean free path through a Dingle analysis (Fig. 3). An upper limit of approximately 200 nm is obtained, which rapidly decreases with increasing temperature. A further scenario based on coherent quantum diffusion in the presence of elastic scattering, in principle, may also reconcile the observation of interference at length scales above a transport mean free path. This scenario at elevated temperatures implies the absence of inelastic scattering despite the substantial phonon density and Fermi–Dirac broadening, which is highly unlikely given the strong electron–phonon coupling in the material[11,37]. Alternatively, we might consider the formation of coherent surface states similar to observations of $h/e$ oscillations in Bi$_2$O$_2$Se nanowires[42]. However, this is difficult to reconcile with the high carrier density of CsV$_3$Sb$_5$ and the global nature of the frequency switching. Thus, coherent propagation of electrons between opposing sidewalls over micrometre distances seems similarly implausible, particularly at temperatures above 20 K.

## Critical ingredient: many-body physics

Single particles do not traverse these samples while retaining phase or velocity information, making it difficult to reconcile the observations with models based on independent single-particle states at the Fermi

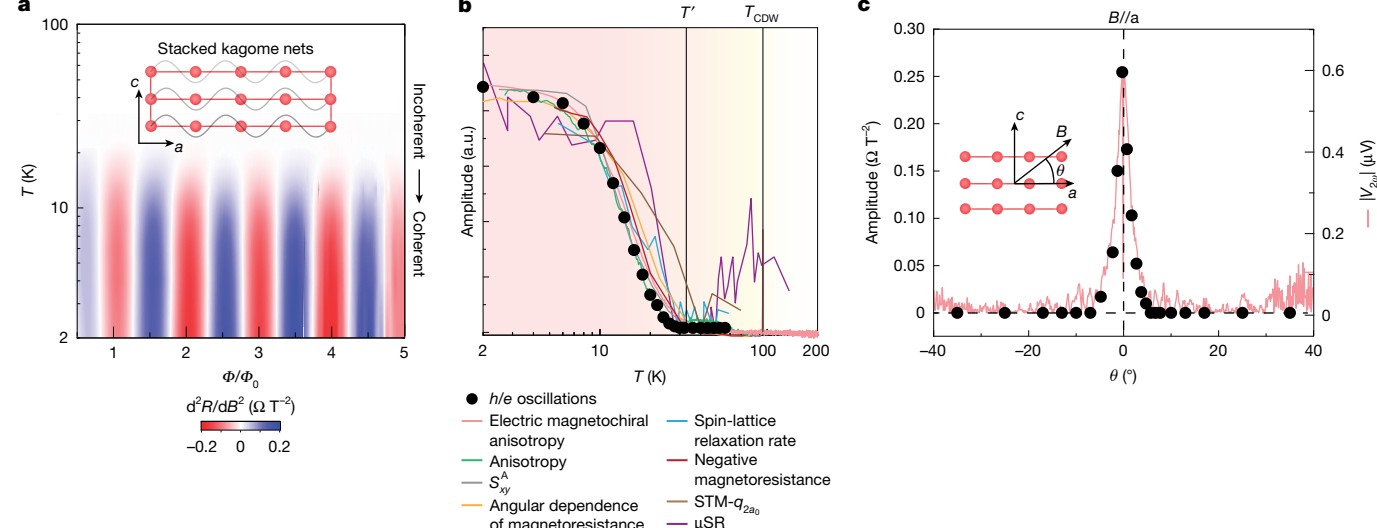

**Fig. 4 | Summary of temperature dependence and comparison with previous reports. a**, $T$ dependence of $h/e$ oscillations in $CsV_3Sb_5$ in device S2. The amplitude becomes prominent below 20 K, suggesting a crossover from quantum incoherent to coherent transport regime. **b**, Summary of the previously reported temperature dependence of complementary experiments, including electric magnetochiral anisotropy[18], field-induced in-plane conductivity anisotropy[33], anomalous Nernst effect[21], angular dependence of magnetoresistance[19], negative magnetoresistance[24], spin-lattice relaxation rate[23], wavevector intensity of the $2a_0$ charge order through STM measurements[47] and muon spin relaxation rate[22]. All results fall on a universal temperature scaling below $T' \approx 30$ K. **c**, The amplitude suppression by out-of-plane field rotation exactly matches that of previously reported switchable chiral transport[18]. a.u., arbitrary units.

level—whether quantum or semiclassical. This motivates the consideration of a coherent many-body state as the underlying mechanism and we now outline the growing body of evidence supporting this interpretation as well as the discrete switching under in-plane field rotation.

First, the angular dependence of the oscillations with respect to the out-of-plane magnetic field direction underscores their distinction from conventional single-particle processes. As the magnetic field is tilted away from the kagome plane ($\theta = 0°$), the oscillations are rapidly suppressed (Fig. 3b, circles). A small misalignment of just $\theta = 5°$ is already sufficient to extinguish them completely. This behaviour contrasts with both semiclassical and quantum single-particle mechanisms, which sense only the projected component of the magnetic flux onto the plane of electronic motion. For such processes, a tilt of 5° has a negligible effect, as $\cos(5°) \approx 0.996$. This expectation is consistent with both theoretical predictions and experimental observations. For example, integration of the semiclassical Bloch–Lorentz equations on the Fermi surface of $CsV_3Sb_5$ as the most advanced semiclassical single-particle model for this system predicts only a mild, continuous modulation of oscillation behaviour over large angular ranges. This prediction aligns with experimental results in other systems, such as Aharonov–Bohm oscillations in $Cd_3As_2$ nanowires[43] and $h/e$ oscillations in $PdCoO_2$ (ref. 38), which show smooth, flux-driven angle dependences. By contrast, a coherent many-body wavefunction that couples to magnetic fields can support singular responses, such as vortex-like rotations of its phase field, as required by the condition of single-valuedness. Notably, the prototypical example for magnetoconductance oscillations from a many-body origin are found in the same experimental configuration of in-plane fields in confined layered superconductors such as cuprates[44]. They can be understood from the commensurability between the integer number of Josephson vortices sandwiched between the superconducting planes and the applied flux. Once vortices are forced to appear within the superconducting plane, driven by tilting the field out of plane, these commensurability oscillations are sharply suppressed—a phenomenology that this system shares with our observations in $CsV_3Sb_5$. We might speculate that vortex-like excitations of the many-body state limit its coherence, providing a natural link to the experimental observations of field-switchable chirality and time-reversal symmetry breaking[31].

It is important to note that magnetoresistance oscillations do not reveal the charge of the fundamental excitation within the many-body state. In principle, quantum fluids composed of constituents with effective charge $q$ exhibit flux periodicities of $h/q$. For instance, Cooper pairs ($q = 2e$) give rise to $h/2e$ oscillations in SQUIDs[2], vortex quantization and Little–Parks effects[3]. However, $h/e$ oscillations in superconducting SNS junctions arise from Andreev interference between electron-like and hole-like trajectories[45]. Likewise, a periodicity of $h/4e$ per layer is observed in intrinsic Josephson junctions hosted by highly anisotropic superconductors such as cuprates and pnictides[46]. Thus, although the observed periodicity reflects the coherence of the underlying state, it does not constrain whether its excitations are elementary or composite nor their effective charge.

Further, their temperature dependence clearly associates the oscillations with a larger modification of the electronic spectrum at $T'$ (Fig. 4). The oscillation amplitude remains nearly constant up to 10 K, followed by a gradual, approximately linear decline into the noise floor around 25 K. This characteristic S-shaped decay closely mirrors the behaviour observed in numerous other experimental probes, including scanning tunnelling microscopy (STM)[47], muon spin rotation (μSR)[22], nuclear magnetic resonance[23], the anomalous Nernst effect[21] and electrical transport[19,20,24]. Although specific heat measurements have found no evidence for a thermodynamic phase transition at $T'$, our data support a scenario in which long-range coherence emerges from a fluctuating, incoherent many-body state formed at higher temperatures, potentially as high as the charge-ordering temperature $T_{CDW}$. This connection is directly reflected in the data, as the suppression of the oscillation amplitude with out-of-plane angle $\theta$ perfectly matches a previous observation of field-switchable chirality[18] (Fig. 4c).

## Conclusions

Taken together, our results support a scenario in which the transition at $T'$ marks the onset of long-range coherence in a many-body state at higher temperatures. Notably, this state is sensitive to the geometry and shape of micron-scale confinement, as well as to the global rotational

symmetry of the three-dimensional crystal structure. Through its phase stiffness, it contributes to charge transport with some parallels to superconductivity, the only other known long-range coherent itinerant state. Although the state remains dissipative and clearly non-superconducting, it exhibits notable similarities to layered superconductors, including magnetoresistance oscillations reminiscent of Josephson plasma coupling in the cuprates[48]. Likely candidates for the incoherent constituents that condense into this coherent state include orbital loop currents, excitons or coupled charge and spin correlations (see Supplementary Information for further discussion). Naturally, explaining the internal structure of this phase responsible for these notable quantum transport signatures requires further studies with local and microscopic probes. Tunnelling microscopy already uncovered key results in this direction, evidencing the formation a coherent unidirectional state in the charge density wave gap at $T'$ (ref. 28). Whether this is a representation of the coherent bulk object itself or a residue of enhanced single-particle coherence as fluctuations of a many-body object condense into an ordered state, it clearly forms the background electronic structure out of which the superconductivity in the system develops[49].

The prospect of coherent charge transport in the normal state of $CsV_3Sb_5$ opens a new frontier in our understanding of correlated quantum matter. These results not only shed light on the enigmatic intermediate-temperature state in kagome metals but also suggest broader design principles for realizing coherence in quantum materials through geometry, frustration or interaction-driven mechanisms. Beyond fundamental insight, the ability to sustain coherent transport without superconductivity may offer new avenues for quantum interference devices operating at elevated temperatures and in regimes in which superconductivity is absent or undesirable. These observations further place strong constraints on microscopic models of the low-energy physics in $CsV_3Sb_5$ and firmly establish kagome metals as a fertile platform for realizing and exploring new coherent electronic states.

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

# Article

## Methods

### Crystal synthesis and device fabrication

Plate-like single crystals of $CsV_3Sb_5$ are obtained following a self-flux procedure as described in ref. 21. They crystallize in the hexagonal structure (*P6/mmm* space group). The microdevices (S1 to S8) are fabricated using the FIB technique (Extended Data Fig. 1). The microstructure fabrication procedure and the initial test on device quality are performed following the previously reported recipe[18,20]. A slab of the bulk material (lamella) is dug out and transferred in situ by a micromanipulator and welded to a gold-coated (Au: 300 nm) $SiN_x$ membrane chip through Pt deposition with ion beams. To clarify the influence of FIB-deposited contacting electrodes, device S3 is gold-coated after the lamella transfer process. The membrane window is later carved into meander-shaped springs, which become the only mechanical connections from the device to the supporting substrate. These soft springs with low spring constant of approximately 150 N m$^{-1}$ substantially reduce the thermal contraction strain. The lamella is then fabricated into the desired shape following a low-voltage (2 kV) and low-current (100 pA) Xe ion beam cleaning as the final fabrication step to reduce the thickness of the FIB-damaged amorphous layer. To reduce the torque force at high magnetic fields, devices S3 and S5 are attached rigidly on one side of the silicon substrate with Pt deposition and the clear consistency among all devices demonstrates the irrelevance of the magnetic torque effect in our measurements.

### Electrical measurements

Resistance measurements were performed using a multichannel SynkTek lock-in in a DynaCool PPMS system with a maximal magnetic field of 14 T. A low AC current of 30 µA is used to measure all devices to avoid Joule heating. The absence of notable self-heating has been confirmed by the absence of higher harmonic responses. The consistent observation of clear Shubnikov–de Haas oscillations and high electric conductivity at low temperatures demonstrates the intact physical properties of all FIB-fabricated devices and the irrelevance of the FIB-damaged amorphous layer, which typically encases the samples with a thickness of 10 nm corresponding to a negligible fraction of the sample cross-section. The field dependence of magnetoresistance at base temperature ($T = 2$ K) shows a slight variation across different devices (Extended Data Fig. 2), mainly attributed to a small misalignment between the magnetic field direction and the kagome plane. The subtracted second derivative demonstrates a consistent oscillation frequency change with varying sample width following the description of $h/e$ oscillations. For an estimation of uncertainty, we fit the width ($w$) dependence of the oscillation period ($\Delta B$) based on the relation $\Phi_0 = \Delta B w c'$. This yields $c' = 9.11 \pm 0.36$ Å, which is nearly identical to the $c$-axis lattice parameter in $CsV_3Sb_5$ ($c = 9.28$ Å). The errors of this analysis are mainly attributed to the uncertainty in determining the exact oscillation period $\Delta B$.

### Subtraction of oscillations: polynomial fitting versus second-order derivative

Two independent methods have been used to remove the background of magnetoresistance to check the consistency of the $h/e$ oscillations subtracted (Extended Data Fig. 3). These methods follow the common practices of the analysis of Shubnikov–de Haas oscillations. First, we directly take the second-order derivative of the field dependence of the magnetoresistance, and the periodic-in-field oscillations are readily observable. When the magnetoresistance has a continuously varying second-order derivative, the resulting background may mask the oscillatory part of the signal. For the complimentary analysis, the magnetoresistance is fitted with a fifth-order polynomial and the subtracted part shows consistent oscillatory behaviour. The polynomial

was checked to ensure that it did not contain notable oscillatory behaviour and did not introduce artificial oscillations into the data. This way, the background is subtracted further, whereas its direct comparison with the second-order derivative reveals a one-to-one correspondence with the π-phase shift. It clearly demonstrates the consistency between these methods in analysing $h/e$ oscillations. All raw data, as well as the background-subtracted data, are made available in the online repository.

### Influence of strain effect

The electronic response in $CsV_3Sb_5$ is extremely sensitive to the influence of strain effect, as consistently demonstrated by previous observations[18,20]. To explore the possible strain-sensitivity of $h/e$ oscillations, we mounted the device directly to a sapphire substrate without the membrane springs for mechanical buffering (Extended Data Fig. 4). The mismatch of the thermal expansion coefficient between the device and the sapphire substrate leads to a tensile strain on the device, which reaches about 0.3% at $T = 2$ K (ref. 20). Its impact is elaborated by the clear increase of the charge-ordering temperature, whereas the broadening of the transition is because of the unavoidable strain inhomogeneity across the microstructure. Notably, the $h/e$ oscillations are strongly suppressed in the strained device. Its comparison with a nearly strain-free device (S6) with a similar sample width demonstrates more than 90% reduction of oscillation amplitude owing to strain. Moreover, the oscillation disappears quickly with increasing field, as only the first period can be clearly resolved. Given that the actual change of lattice parameter owing to tensile strain is less than 0.5%, the deconstructive interference caused by the inhomogeneous lattice constant across the strained device cannot be the main source of the reduction observed. By contrast, it aligns with the observed suppression of strain in magnetochiral transport signatures[18]. This indicates a fundamental link between long-range coherence and correlated electronic order in $CsV_3Sb_5$.

## Data availability

The data supporting the findings of this study are deposited on Zenodo, https://doi.org/10.5281/zenodo.17067846 (ref. 50). Source data are provided with this paper.

## Code availability

The code supporting the findings of this study are deposited on Zenodo, https://doi.org/10.5281/zenodo.17067846 (ref. 50).

50. Guo, M. Many-body interference in Kagome crystals. *Zenodo* https://doi.org/10.5281/zenodo.17067845 (2025).

**Acknowledgements** We thank J. P. Ingham, R. Thomale, H. Scammell, R. M. Fernandes, A. Stern and F. von Oppen for the insightful discussions. This work was supported by the European Research Council (ERC) under grant Free-Kagome (grant agreement no. 101164280). This work was supported in part by the Deutsche Forschungsgemeinschaft under grant SFB 1143 (project-id 247310070), the cluster of excellence ct.qmat (EXC 2147, project-id 390858490) and the European Research Council (ERC advanced grant no. 742068 'TOPMAT'). This work was supported by the Swiss National Science Foundation through a Consolidator Grant (iTQC, TMCG-2_213805). This work was supported by HFML-RU/NWO-I, a member of the European Magnetic Field Laboratory (EMFL). Part of this work was performed at the National High Magnetic Field Laboratory, which is supported by National Science Foundation Cooperative Agreement no. DMR-2128556, the State of Florida and the US Department of Energy. R.D.M. is supported through the Center for the Advancement of Topological Semimetals (CATS), an Energy Frontier Research Center (EFRC) funded by the US Department of Energy (DOE) Office of Science, under contract DE-AC0207CH11358.

**Author contributions** Crystals were synthesized and characterized by D.C. and C.F. The experiment design, FIB microstructuring and magnetotransport measurements were performed by C.(M.)G., L.Z., C.P. and P.J.W.M. M.R.v.d. and S.W. assisted with the high-field magnetotransport measurements at the 35-T static magnet and F.F.B. and R.M. helped with the electrical measurements in the pulsed magnet up to 65 T. K.W. performed the

theoretical simulation based on the semiclassical Bloch–Lorentz model and the density functional theory calculations were performed by M.G.-A., M.A., I.E. and M.G.V. The experimental results were analysed by C.(M.)G. and P.J.W.M. All authors contributed to the writing of the paper.

**Funding** Open access funding provided by Max Planck Society.

**Competing interests** The authors declare no competing interests.

**Additional information**
**Correspondence and requests for materials** should be addressed to Chunyu Guo (Mark) or Philip J. W. Moll.

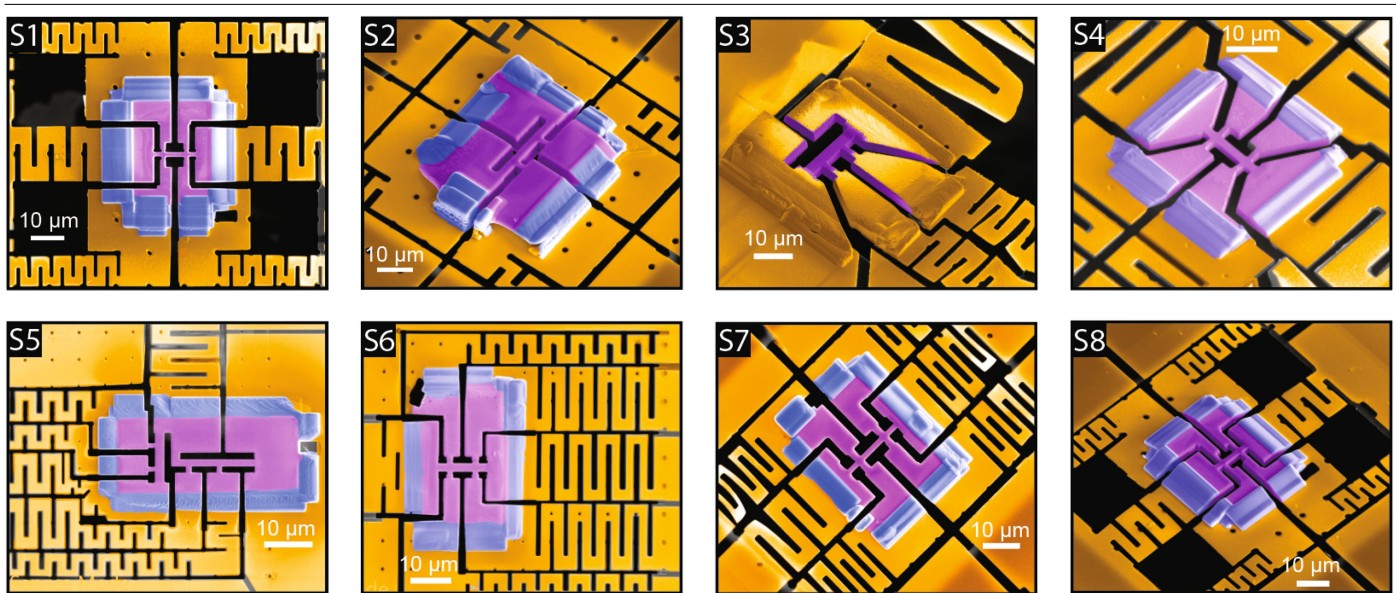

**Extended Data Fig. 1 | Scanning electron microscope images of all devices.** All microstructures use the soft membrane springs to reduce the thermal differential strain effect at low temperatures.

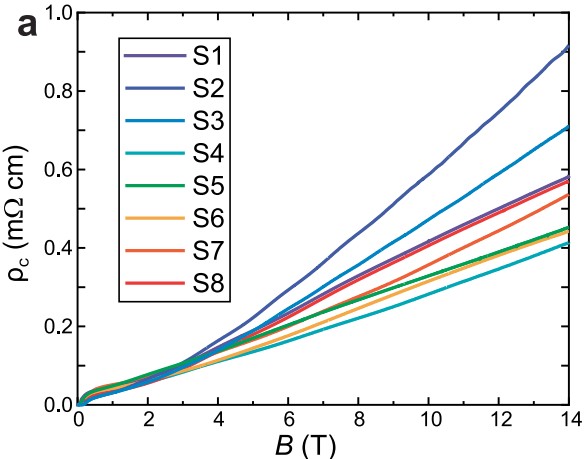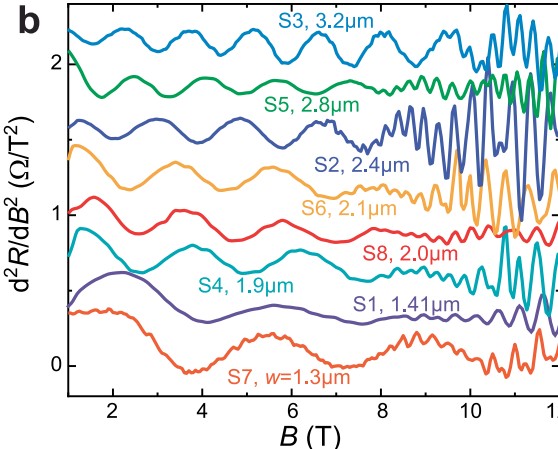

**Extended Data Fig. 2 | Summary of all magnetoresistance and corresponding $h/e$ oscillations. a**, The magnetoresistance of all devices increases monotonically with increasing field and the difference between their values is mainly attributed to the inevitable misalignment between the magnetic field and crystalline $a$ direction. **b**, The second derivative of the field-dependent magnetoresistance clearly reveals the presence of $h/e$ oscillations in all devices. Note that each curve is shifted by at least $0.3\ \Omega\ T^{-2}$ for clarity.

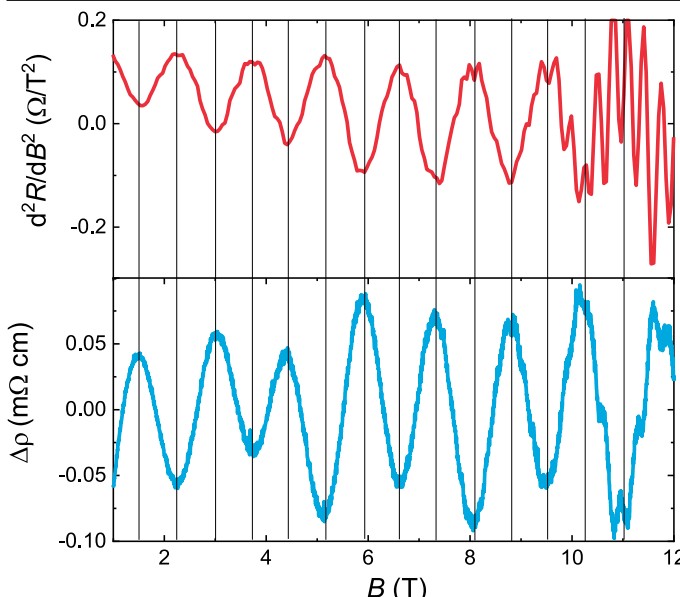

**Extended Data Fig. 3 | Comparison of *h*/*e* oscillations subtracted by two different methods.** The *h*/*e* oscillations can be clearly resolved by either taking the second field derivative or subtracting the fifth-polynomial fitting as a background. The main difference between these two methods is a π-phase shift, as demonstrated by the flipped peak–valley correspondence between them.

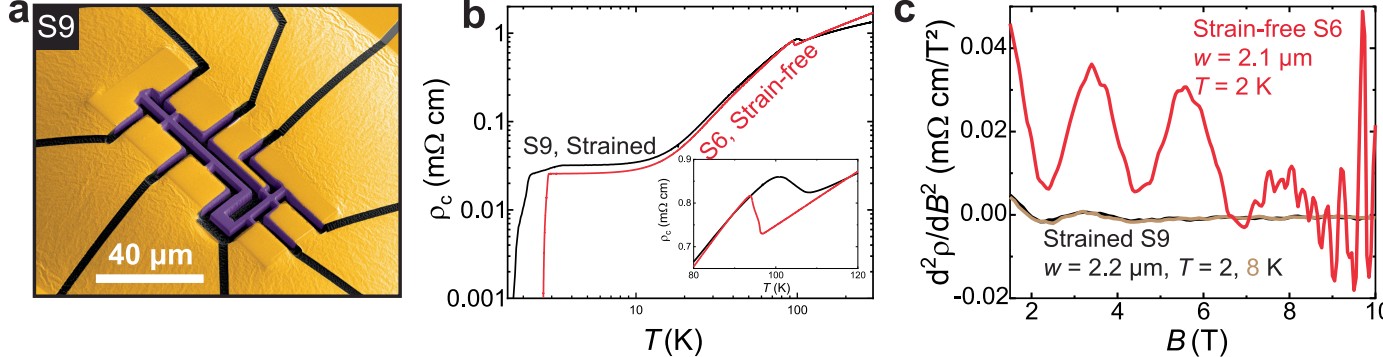

**Extended Data Fig. 4 | Influence of uniaxial strain. a,** Scanning electron microscope image of device S9. The device is mechanically attached to a sapphire substrate by means of a double-component glue droplet. At low temperatures, the thermal expansion coefficient mismatch between the device and the substrate results in a substantial uniaxial strain across the device. **b,** $T$-dependent resistivity of strained and nearly strain-free devices. Owing to the presence of tensile strain, the charge-order temperature is clearly increased and its broadening is possibly attributed to the strain inhomogeneity. **c,** Comparison of $h/e$ oscillations in both devices. The oscillations are strongly suppressed in the strained device and only one oscillation period is visible. This is expected given the extraordinary strain-sensitivity of $CsV_3Sb_5$, as consistently demonstrated in previous works[18,20].