## [Peer Review file · Nature]

Many-body interference in Kagome crystals

Corresponding Author: Dr Chunyu Guo

Version 0:

Reviewer comments:

Referee #1

(Remarks to the Author)

In this work, Guo et al. reported the magnetoresistance oscillation, which is periodic in B , in mesoscopic pillars of the kagome metal CsV_3Sb_5 under an in-plane magnetic field, and attributed this phenomenon to the enigmatic phase coherence of mobile electrons on the kagome plane. Their experimental finding appears interesting. However, similar oscillations have also been reported in the quasi-2d PdCoO_2 family [Ref. 4]; No acceptable explanations of the oscillation itself and its angle dependence have been proposed; Despite the quantitative agreement of its temperature dependence with previous STM, μSR and many other probes, no further insight or information has been provided to understand the intermediate temperature scale T^* . As a result, I don't think the impact and importance of this work have met the criterial of Nature.

Below are some other questions:

1. The authors seem to mix the mean free path and the phase coherent length. They spent many paragraphs on the estimation and discussion of the former, which is entirely different from the phase coherent length central to the oscillation. These two scales can vary: Actually in PRB 103,045123 (2021), the estimated phase coherent length of PdCoO_2 device is 10 times larger than its mean free path. Current sample size is apparently not enough.
2. The claimed abrupt discontinuous oscillation period at 45 deg is weird in Fig.3b. Maybe the data is oversmoothed, or the two branches just smeared together in the FFT analyses in Fig.S11. Usually FFT analysis for weak, long-period oscillation in limited window is not reliable.

Referee #2

(Remarks to the Author)

I co-reviewed this manuscript with one of the reviewers who provided the listed reports.

Referee #3

(Remarks to the Author)

The manuscript by Guo et al. reports the observation of magnetoresistance oscillations that are periodic in magnetic field in micrometer-sized rectangular bars of CsV_3Sb_5 , which are attributed to long-range electron coherence. While both the measured phenomenon and charge-ordered kagome metals are independently interesting, we do not find that this work meets the high bar of Nature. This is primarily due to two reasons:

1. The periodic-in-field magnetoresistance oscillations reported here have already been observed in $(\text{Pd,Pt})\text{CoO}_2$, which diminishes the novelty of this work. Although the authors highlight differences between CsV_3Sb_5 and $(\text{Pd,Pt})\text{CoO}_2$, they provide no clear explanation for how this phenomenon could occur in two such distinct systems.
2. Lack of mechanistic insight: Since no clear understanding of these periodic-in-field oscillations is established, the authors resort to comparing measurements on CsV_3Sb_5 using various electronic probes, all of which exhibit a similar temperature dependence (Fig. 4b). This plot is somewhat unconvincing, particularly on a linear-log scale. Moreover, this underscores the absence of new fundamental understanding regarding charge-ordered kagome metals, as the authors fail to connect these probes meaningfully and instead merely add another to the existing list of phenomena with comparable temperature dependence.

Additional Points:

- The authors simulate semiclassical Bloch-Lorentz oscillations using a simple quasi-2D Fermi surface. However, the Fermi surface of CsV₃Sb₅ is far more complex, especially considering charge density wave (CDW) reconstruction. Quasi-2D pockets near the van Hove singularity—formed by bands touching the Brillouin zone boundary (see, e.g., Phys. Rev. X 11, 041030 (2021))—could produce open orbits sensitive to field direction, potentially explaining the observed angle dependence. Ruling out Bloch-Lorentz oscillations without accounting for the realistic Fermi surface of CsV₃Sb₅ seems premature.
- On page 6, the authors state: “On the contrary, it again demonstrates a non-analytical response to a magnetic field of the electronic system in CsV₃Sb₅, in accordance with other experimental probes [20].” However, while they compare their in-plane angle dependence to prior chiral transport results, no similar in-plane angle dependence is evident in the cited study.
- Could the in-plane angle dependence be explained by considering the magnetic field projection switching from top/bottom surfaces to side surfaces at 45 degrees?
- The out-of-plane angle dependence of the periodic-in-field oscillations is plotted alongside chiral magnetotransport in Fig. S8b. Since this comparison is noted in the main text, it should be included in Fig. 2b for clarity.
- Some figure references in the supplement are incorrect. For example, Section H erroneously cites Fig. S8.

Referee #4

(Remarks to the Author)

I co-reviewed this manuscript with one of the reviewers who provided the listed reports.

Referee #5

(Remarks to the Author)

In this manuscript, “Long-range electron coherence in Kagome metals”, the authors report the observation of h/e-periodic magnetoresistance oscillations in the Kagome metal CsV₃Sb₅, occurring at temperatures above 20 K and in micron-scale devices where single-particle mean free paths are significantly shorter than the device dimensions. This challenges conventional wisdom about coherence in metals, as such effects are typically restricted to sub-Kelvin temperatures and ballistic regimes.

The findings in this work are original, as they extend h/e oscillations to a new material class (Kagome metals) with distinct correlated phenomena. While the effect in (Pd,Pt)CoO₂ was attributed to ballistic transport in high-purity oxides, the present work shows coherence in a disordered, multiband system, implying a fundamentally different origin tied to many-body interactions. This addresses an open question in condensed matter physics: how to achieve long-range coherence in metals beyond single-particle pictures.

While the manuscript identifies that single-particle coherence (ballistic transport) cannot explain the observations, the proposed collective mechanism (e.g., loop currents, excitonic order) remains speculative. These experimental findings would strongly motivate further theoretical studies.

This manuscript highlights a material platform where interaction-stabilized long-range coherence persists despite strong scattering. This is crucial for fundamental studies of quantum interference in metals and potential technological applications, thus should be interested to a broad readership.

In conclusion, I would recommend publication in Nature after the authors properly address the following concern..

- 1) The h/e-period magnetoresistance oscillations were also reported in Bi₂O₂Se nanowires [Phys. Rev. B. 100, 235307 (2019)], in addition to Delafossite oxides (PtCoO₂, PdCoO₂) [Science 368, 1234 (2020)]. Though the Bi₂O₂Se nanowires are quasi-1D semiconducting nanostructures with gate-tunable surface states, distinctly different to the bulk CsV₃Sb₅ kagome metal studied in this manuscript, it would be desirable if the authors can comment on the h/e oscillations in Bi₂O₂Se nanowires.
- 2) STM see the quasiparticle interference pattern originated from the scattering among Fermi surfaces below T' ~ 30-35 K [see for example, Nat. Phys. 19, 637 (2023) and Nature 632, 775 (2024)], while it disappears above T', leaving quasiparticle interference peaks at some disconnected wave vectors. I wonder if it has anything to do with the long-range electron coherence the authors addressed in this manuscript. It would be great if the authors can comment on the STM observation.
- 3) The caption for Fig. 3b is kind of confusing to me. It says that “the configuration of in-plane rotation and its correspondence to the device’s cross-section are illustrated in the left-hand inset”. However, what I saw are four insets displaying R as a function of B at four in-plane angles.
- 4) There seems to have some typos in the sentence “Further increasing the angle towards the surface normal of the other sidewalls ...” in the second paragraph in the subsection ‘Exotic angular dependence’. The effective width should now correspond to the wider sidewall with w=2.89 μm, and the data are displayed by the purple branch in Fig. 3.

Version 1:

Reviewer comments:

Referee #1

(Remarks to the Author)

In the revised manuscript, the authors have made some modifications by adding experimental data of devices with parallelogram-shaped cross-sections and the analyses on the quantum mean free path. They argue that their results gave experimental evidence for global electronic coherence and a field-angle-dependent switching between distinct quantum states incompatible with single-particle description, which distinct the present work from PdCoO₂ and adding new piece to

the previous identified T' phase in CsV3Sb3. To reach these conclusion (and to distinguish the CsV3Sb5 system from the phase coherence scenario in previous PdCoO2 family), the authors have stressed on the “quantum mean free path” derived from the Dingle temperature, which, however, is entirely different from the true phase coherence length as shown below: The transport mean free length L_m , the quantum scattering length (L_q , derived from Dingle temperature, or quantum mean free path denoted by the authors in the rebuttal) and the phase coherence length (L_ϕ) depict entirely different scattering processes:

1. L_m is the average distance an electron travels between any scattering event (either elastic or inelastic), and is somehow insensitive to small-angle scattering (only large-angle scattering contributes strongly). This value is weakly temperature-dependent.

2. L_q is the average distance an electron travels before its quantum phase is randomized by elastic scattering. Corresponding scattering time accounts for all elastic scattering events regardless of its scattering angle. Note this is also the reason why L_q is consistently much smaller than L_m in most reported systems including the CsV3Sb5 here, as small-angle scattering is usually common in materials.

3. L_ϕ is the maximum distance an electron retains quantum phase coherence, which is governed by inelastic (e.g., e-e or e-p) scattering. This value is strongly temperature-dependent. At low temperatures, L_ϕ can be significantly larger than L_q , L_m or even the sample size, as in the case of 2D electron gas.

The obfuscation of L_q and L_ϕ , and the resultant wrong estimation of true phase coherence length in the article severely limit the scientific soundness, making the critical declaration on page 7 (“Thus, the phase coherence length may well exceed the transport m.f.p., yet this does not hold for CsV3Sb5”) hold no water. Moreover, a microscopic model remains elusive, and the proposed many-body related physics have been frequently reported by previous STM and many other probes (Refs.46,20,21,22, etc.).

As a result, I do not see sufficient soundness and depth of the study to justify the publication in Nature. Nevertheless, the experimental part of this work is interesting and certainly suited for a high-impact journal on lower general importance level than Nature after necessary changes.

Referee #2

(Remarks to the Author)

I co-reviewed this manuscript with one of the reviewers who provided the listed reports.

Referee #3

(Remarks to the Author)

The updated manuscript by Guo et al. addresses many of the issues raised by reviewers in the first round. Of particular importance, this draft highlights the key differences between the observation of periodic in magnetic field magnetoresistance oscillations in CsV3Sb5 and (Pd,Pt)CoO2. While it is still mechanistically unknown what is driving these oscillations in CsV3Sb5, with the new data presented here on parallelogram-shaped bars it is clear that this result cannot be understood by a simple independent single-particle state picture. As such, we recommend the publication of this work in Nature.

One important typo that needs to be fixed: at the bottom of page 5 a part of a sentence seems to be missing: “A striking result emerged: while the large oscillations again follow the same $\cos(\phi) - 1$ scaling, set by the different widths of both sidewalls(Fig. 2d).”

Referee #4

(Remarks to the Author)

I co-reviewed this manuscript with one of the reviewers who provided the listed reports.

Referee #5

(Remarks to the Author)

The authors have provided thorough answers to my questions in the last review, where they have included much more detailed discussions and analyses. I recommend the publication of this manuscript on Nature.

Dear Reviewers,

We sincerely thank you for your careful reading of the manuscript and your insightful comments. The review process has been invaluable in revealing that the original submission failed to convey our central message clearly. In response, we have completely rewritten the manuscript to emphasize its main conclusion: **evidence of a macroscopically coherent many-body electronic state contributing to charge transport distinct from superconductivity.**

The magnetoresistance oscillations unmistakably reveal a fundamentally new regime of coherent charge dynamics. This conclusion rests on a series of reproducible experiments that cannot be explained by either semiclassical or fully quantum single-particle theories, providing critical input to the development of a microscopic theory for the T' state.

We are especially grateful for the suggestions regarding further experiments. These have led to new, compelling results. In particular, devices with parallelogram-shaped cross-sections exhibit distinct modifications of the oscillation pattern, reflecting a global geometric influence on the emergent state. Most strikingly, we consistently observe a single oscillation frequency and robust discrete switching, reinforcing the picture of a confined, geometry-defined energy landscape for the T' phase.

Before addressing the detailed referee comments, we wish to respond to two overarching concerns raised by all reviewers:

(1) Apparent similarity to PdCoO₂:

We found the parallel to PdCoO₂, raised uniformly across the reviews, to be particularly revealing. In our view, the resemblance is superficial and misleading—something the original manuscript did not sufficiently clarify. All quantum mechanical systems of electrons fundamentally oscillate at a period given by h/q , with q the effective charge of the elementary excitation. Any oscillatory phenomenon based on the orbital motion of quantum mechanical electrons follows this principle: ($q = e$) Aharonov-Bohm oscillations, Hofstadter patterns, and de Haas van Alphen oscillations (*the fundamental process is periodic in $B^{-1} = \frac{h}{e} r_c^2$ due to the dual role of the magnetic field setting the cyclotron area and simultaneously the flux through it*), ($q = 2e$) sets the periodicity of superconducting phenomena such as SQUID oscillations, Little-Parks oscillations, etc. However, it is not possible to uniquely determine the charge from the period directly, as it depends on the microscopic details about how the interferometer is formed. For example, single-particle processes such as Aharonov-Altshuler oscillations can oscillate at $h/2e$ while superconductor-metal-superconductor Josephson junctions oscillate at h/e due to the phase difference of electrons and holes in the Andreev process.

What distinguishes our system is that none of the single-particle paradigms applicable to PdCoO₂ can account for the observations in CsV₃Sb₅, particularly given that the device dimensions far exceed the single-particle mean free path. Instead, our results reflect a collective, long-range coherent response—a hallmark of electronic order, previously only known from superconductors.

(2) Absence of a microscopic model for the T' phase:

We agree that the microscopic origin of the T' phase remains elusive. Nevertheless, we provide clear experimental evidence for global electronic coherence and a field-angle-dependent switching between distinct quantum states—features incompatible with any known single-particle description. This adds a critical piece to the puzzle of CsV₃Sb₅: coherence.

While superconductivity and charge order in this system increasingly appear conventional and phonon-driven, the transition at T'—until now considered a subtle spectral anomaly—reveals itself here as a

defining boundary of a novel coherent state. This state is intimately connected to the emergence of switchable electronic chirality, one of the most enigmatic properties of this material.

We fully share the reviewers' desire to uncover the microscopic mechanism. In the supplement, we outline several plausible scenarios including loop currents, unconventional density waves, and two-fluid models. We further provide stringent experimental constraints for future theoretical work. Our data establish a robust phenomenology across variations in sample size, shape, field orientation, temperature, and disorder—laying the foundation for identifying the true nature of this coherent electronic state.

Referee #1 (Remarks to the Author):

In this work, Guo et al. reported the magnetoresistance oscillation, which is periodic in B , in mesoscopic pillars of the kagome metal CsV_3Sb_5 under an in-plane magnetic field, and attributed this phenomenon to the enigmatic phase coherence of mobile electrons on the kagome plane. Their experimental finding appears interesting. However, similar oscillations have also been reported in the quasi-2d PdCoO_2 family [Ref. 4]; No acceptable explanations of the oscillation itself and its angle dependence have been proposed; Despite the quantitative agreement of its temperature dependence with previous STM, μSR and many other probes, no further insight or information has been provided to understand the intermediate temperature scale T' . As a result, I don't think the impact and importance of this work have met the criterion of Nature.

We hope that the revised manuscript, together with the new and rather unexpected experimental results, addresses these core concerns. At the heart of this discovery lies precisely the fact that existing theoretical frameworks—particularly those developed for PdCoO_2 —fail to describe or anticipate the observed phenomena. Our central contribution is the identification of a coherent many-body state, supported by a comprehensive and reproducible set of experimental signatures defining its phenomenology.

These findings have already sparked significant interest in the theory community, prompting diverse and independent efforts to propose viable microscopic models and to predict new experimental observables. At this early stage, it would be premature to commit to any single theoretical framework. Instead, we outline the leading conceptual directions currently under exploration, which our experimental work strongly constrains and will help guide us going forward.

Below are some other questions: 1. The authors seem to mix the mean free path and the phase coherent length. They spent many paragraphs on the estimation and discussion of the former, which is entirely different from the phase coherent length central to the oscillation. These two scales can vary: Actually in PRB 103,045123 (2021), the estimated phase coherent length of PdCoO_2 device is 10 times larger than its mean free path. Current sample size is apparently not enough.

The revised manuscript places even greater emphasis on the crucial distinction between the phase coherence length (here referred to as the quantum mean free path) and the transport mean free path. While any quantum process can exhibit oscillations with a period of h/e , this alone does not establish phase coherence across the device, which is defined as the ability of an electronic wavefunction to interfere with itself. Importantly, semi-classical theories are also quantum mechanical and thus can also yield h/e oscillations without requiring phase coherence, as demonstrated by Vilkelis *et al.* (SciPost Phys. 15, 019 (2023)). However, such descriptions are only valid when the sample size remains much smaller than the transport mean free path, making it essential to consider both lengths independently.

These two quantities—phase coherence length and transport mean free path—are generally distinct and can differ by orders of magnitude. We therefore treat them separately. The quantum mean free

path is extracted from standard Dingle analysis of quantum oscillations, yielding values significantly smaller than the device width, speaking against single-particle quantum interference as the origin of the observed phenomena. This methodology has proven reliable in PdCoO₂, where the results of Dingle analysis align well with observed universal quantum fluctuations (PRB 103, 045123 (2021)). In nanostructured devices, the somewhat reduced phase coherence length (~100 nm) compared to single crystals (~400 nm, Putzke *et al.*, Science 368, 1234 (2020)) is consistent with higher defect densities—particularly twin boundaries—which are known to limit phase coherence.

Thus, both transport and quantum mean free paths are central to this discussion; yet, both are far smaller than the device dimensions, thereby excluding any single-particle interpretation. The revised manuscript now treats these considerations with explicit and separate discussions.

2.The claimed abrupt discontinuous oscillation period at 45 deg is weird in Fig.3b. Maybe the data is oversmoothed, or the two branches just smeared together in the FFT analyses in Fig.S11. Usually FFT analysis for weak, long-period oscillation in limited window is not reliable.

While we fully agree that frequency analysis presents inherent challenges, particularly in the presence of abrupt spectral changes, we are puzzled by this specific critique. The potential limitations of Fourier analysis and its inability to cleanly resolve nearby frequencies in limited field windows were explicitly discussed in the manuscript. For this reason, we deliberately based our interpretation not on Fourier transforms, but on the raw experimental data, as shown in the main figure insets (reproduced below for clarity). This is raw data without smoothing beyond the 2s time constant of the lock-in amplifier during the slow field sweeps. A smooth polynomial was subtracted to treat the background. That this procedure correctly represents higher frequencies is self-evident from the appearance of high-frequency SdH oscillations in the high-field region.

Fig.R1 Reproduced angular dependence of oscillation periods in device S1, see also Fig.2b in the manuscript. The insets display h/e oscillations measured at around the switching angle, and the clear distinction between them demonstrates the critical switching behavior unambiguously.

The discrete and discontinuous shift in oscillation period—with preserved amplitude—is clearly visible in the raw traces and cannot be attributed to analysis artifacts. This abrupt switching is not only visually apparent but also reproducibly observed across multiple devices and geometries. These features are central to our interpretation, as they strongly support the presence of a switchable, coherent many-body state, a conclusion that cannot be accounted for by single-particle interference mechanisms.

Referee #3 (Remarks to the Author):

The manuscript by Guo et al. reports the observation of magnetoresistance oscillations that are periodic in magnetic field in micrometer-sized rectangular bars of CsV_3Sb_5 , which are attributed to long-range electron coherence. While both the measured phenomenon and charge-ordered kagome metals are independently interesting, we do not find that this work meets the high bar of Nature. This is primarily due to two reasons:

1. The periodic-in-field magnetoresistance oscillations reported here have already been observed in $(\text{Pd,Pt})\text{CoO}_2$, which diminishes the novelty of this work. Although the authors highlight differences between CsV_3Sb_5 and $(\text{Pd,Pt})\text{CoO}_2$, they provide no clear explanation for how this phenomenon could occur in two such distinct systems.
2. Lack of mechanistic insight: Since no clear understanding of these periodic-in-field oscillations is established, the authors resort to comparing measurements on CsV_3Sb_5 using various electronic probes, all of which exhibit a similar temperature dependence (Fig. 4b). This plot is somewhat unconvincing, particularly on a linear-log scale. Moreover, this underscores the absence of new fundamental understanding regarding charge-ordered kagome metals, as the authors fail to connect these probes meaningfully and instead merely add another to the existing list of phenomena with comparable temperature dependence.

Thank you for these comments, which we have addressed in the opening statement in depth.

The context of the critique on temperature dependence remained somewhat vague, making it difficult to respond concisely to the point. The plot shows substantial correlations between signals measured by vastly different probes, from μSR to STM, which collectively exhibit signatures of a substantial modification of the electronic spectrum, in the absence of a specific heat anomaly and hence a phase transition in the thermodynamic sense. This correlation is evident in the data, regardless of the chosen scale, while naturally, there is scatter between the data obtained from various crystals using different techniques. Assuming the critique pertains to the use of a logarithmic scale, we present it here on a double-linear scale for comparison. We felt that the log scale was more appropriate, as it shows that most signals are zero at the charge density and above, remain zero at a substantial temperature below T_{CDW} , and then rise in a common S-shape.

Fundamentally, we share your concern about the visual impression of data representation, particularly through the thoughtful choices of scales and axes. We had chosen this scale to improve clarity, not to hide discrepancies. If we miss a salient aspect of the data representation and inadvertently mislead the reader, we apologize and are happy to adjust to a fairer representation.

Fig.R2 Comparison between the log-scale and linear scale representation of the summary of experimental results measured with various probes.

Additional Points: • The authors simulate semiclassical Bloch-Lorentz oscillations using a simple quasi-2D Fermi surface. However, the Fermi surface of CsV₃Sb₅ is far more complex, especially considering charge density wave (CDW) reconstruction. Quasi-2D pockets near the van Hove singularity—formed by bands touching the Brillouin zone boundary (see, e.g., Phys. Rev. X 11, 041030 (2021))—could produce open orbits sensitive to field direction, potentially explaining the observed angle dependence. Ruling out Bloch-Lorentz oscillations without accounting for the realistic Fermi surface of CsV₃Sb₅ seems premature.

In response, we have now performed a calculation on a realistic Fermi surface matching DFT calculations. The angle dependence still fails to describe the experimental results. This is not surprising, as to first order this rotation only reduces the Lorentz force until the field-driven orbits are substantially changed. However, the Fermi surfaces are rather smooth, well-behaved objects, with only minor changes in orbital structure within a few degrees. Geometrically, it would not have been possible to achieve such a substantial suppression by tilting the field by a few degrees. Yet we fully agree that this increases the confidence in the results and eliminates a potential doubt.

Fig.R3 Semiclassical simulation based on realistic Fermi surfaces. The angular dependence of the simulated Bloch-Lorentz oscillations persists to show a clear distinction compared to the experimental results.

• On page 6, the authors state: *‘‘On the contrary, it again demonstrates a non-analytical response to a magnetic field of the electronic system in CsV₃Sb₅, in accordance with other experimental probes [20].’’* However, while they compare their in-plane angle dependence to prior chiral transport results, no similar in-plane angle dependence is evident in the cited study.

Indeed, the in-plane angle study is a novel aspect of this paper. However, the non-analytical dependence of the chirality on the out-of-plane fields is the core of ref [16]. A switch between two states of chirality $\chi(B) \propto \text{sign}(B_z)$ is observed, which clearly classifies as ‘‘a non-analytical response to a magnetic field of the electronic system in CsV₃Sb₅’’. We hope this point is now discussed better in the revised version.

Fig.R4 **a**, Field dependence of chiral conductivity $\Delta\sigma$ at various angles. Between $\vartheta = \pm 0.3$ and $\pm 4.8^\circ$ all data are measured with an angle step of 0.5° . **b**, Angular dependence of the first-order derivative $\partial(\Delta\sigma)/\partial B$ from $B = 6$ to 18 T. The green curve represents the model description of chiral conductivity.

- Could the in-plane angle dependence be explained by considering the magnetic field projection switching from top/bottom surfaces to side surfaces at 45 degrees?

We highly appreciate this comment, which, in synchrony with the other reviewers, inspired the new and surprising experiments on parallelogram-shaped bars. They conclusively demonstrate that there is always only one frequency present, excluding the possibility that each surface switches on independently when the projection exceeds 45 degrees. This was a well-placed hypothesis; however, the response is even more striking, and the simultaneous switching clearly evidences one bulk switching event rather than things happening independently on various surfaces or regions of the sample. This excellent question should be fully resolved in the light of the new data. Thank you.

- The out-of-plane angle dependence of the periodic-in-field oscillations is plotted alongside chiral magnetotransport in Fig. S8b. Since this comparison is noted in the main text, it should be included in Fig. 2b for clarity.

Thank you for this good suggestion, we have adapted Fig. 4c accordingly.

- Some figure references in the supplement are incorrect. For example, Section H erroneously cites Fig. S8.

Thank you, we fixed this.

Referee #5 (Remarks to the Author):

In this manuscript, "Long-range electron coherence in Kagome metals", the authors report the observation of h/e -periodic magnetoresistance oscillations in the Kagome metal CsV_3Sb_5 , occurring at temperatures above 20 K and in micron-scale devices where single-particle mean free paths are significantly shorter than the device dimensions. This challenges conventional wisdom about coherence in metals, as such effects are typically restricted to sub-Kelvin temperatures and ballistic

regimes.

The findings in this work are original, as they extend h/e oscillations to a new material class (Kagome metals) with distinct correlated phenomena. While the effect in (Pd,Pt)CoO₂ was attributed to ballistic transport in high-purity oxides, the present work shows coherence in a disordered, multiband system, implying a fundamentally different origin tied to many-body interactions. This addresses an open question in condensed matter physics: how to achieve long-range coherence in metals beyond single-particle pictures.

While the manuscript identifies that single-particle coherence (ballistic transport) cannot explain the observations, the proposed collective mechanism (e.g., loop currents, excitonic order) remains speculative. These experimental findings would strongly motivate further theoretical studies. This manuscript highlights a material platform where interaction-stabilized long-range coherence persists despite strong scattering. This is crucial for fundamental studies of quantum interference in metals and potential technological applications, thus should be interesting to a broad readership. In conclusion, I would recommend publication in Nature after the authors properly address the following concern.

We thank you for the thorough and substantive assessment of our work, which focuses precisely on the manuscript's main strengths and weaknesses.

1) The h/e-period magnetoresistance oscillations were also reported in Bi₂O₂Se nanowires [Phys. Rev. B. 100, 235307 (2019)], in addition to Delafossite oxides (PtCoO₂, PdCoO₂) [Science 368, 1234 (2020)]. Though the Bi₂O₂Se nanowires are quasi-1D semiconducting nanostructures with gate-tunable surface states, distinctly different to the bulk CsV₃Sb₅ kagome metal studied in this manuscript, it would be desirable if the authors can comment on the h/e oscillations in Bi₂O₂Se nanowires.

Thank you for pointing out this interesting reference. We have added a statement to the paper. It adds the interesting angle of surface quantization. Indeed, in Bi₂O₂Se, surface band quantization is at the heart of the oscillation. Given the high carrier density of CsV₃Sb₅, it is unlikely to be at the origin, in addition to its inherent insensitivity to the interlayer dimension in our experiment on transverse magnetoresistance (as compared to the longitudinal configuration considered in Bi₂O₂Se).

2) STM see the quasiparticle interference pattern originated from the scattering among Fermi surfaces below $T^* \sim 30\text{-}35\text{ K}$ [see for example, Nat. Phys. 19, 637 (2023) and Nature 632, 775 (2024)], while it disappears above T^* , leaving quasiparticle interference peaks at some disconnected wave vectors. I wonder if it has anything to do with the long-range electron coherence the authors addressed in this manuscript. It would be great if the authors can comment on the STM observation.

We appreciate this comment and have added a discussion to the concluding section of the paper. Indeed, these STM results showing quasiparticle coherence and their impact on the superconducting state are most relevant to the story. Clearly, a more substantive discussion, including the reevaluation of STM data, is necessary. Especially the coherent 1D mode is most relevant, yet its coherence length appears limited (it was not evaluated quantitatively, but judging from the FWHM of the peaks it is in the 10nm range). Our view is that this increase in coherence is likely a symptom, not a cause, as in the disappearance of fluctuations associated with the many-body state, boosting the coherence of the residual quasiparticles – a thought guided by the coherence of nodal excitations in d-wave superconductors. While an intriguing speculative thought, this needs to be grounded on more rigorous and microscopic principles, which is why we would avoid overstressing it in the paper.

3) The caption for Fig. 3b is kind of confusing to me. It says that “the configuration of in-plane rotation

and its correspondence to the device's cross-section are illustrated in the left-hand inset". However, what I saw are four insets displaying ΔR as a function of B at four in-plane angles.

4) There seems to have some typos in the sentence "Further increasing the angle towards the surface normal of the other sidewalls ..." in the second paragraph in the subsection 'Exotic angular dependence'. The effective width should now correspond to the wider sidewall with $w=2.89 \mu\text{m}$, and the data are displayed by the purple branch in Fig. 3.

Thank you for pointing these out. In retrospect, these were indeed not formulated well. We hope that the revised manuscript and caption will make the paper much clearer.

We would like to thank you again for your in-depth assessment of the work and hope you share our excitement for the new parallelogram experiments.

Dear Reviewers,

We deeply appreciate your thorough review of the manuscript and your valuable feedback. The review process has been crucial in improving the quality and clarity of the present manuscript. Here we will address the detailed comments/suggestions by the referees.

Referee #1 (Remarks to the Author):

In the revised manuscript, the authors have made some modifications by adding experimental data of devices with parallelogram-shaped cross-sections and the analyses on the quantum mean free path. They argue that their results gave experimental evidence for global electronic coherence and a field-angle-dependent switching between distinct quantum states incompatible with single-particle description, which distinct the present work from PdCoO₂ and adding new piece to the previous identified T' phase in CsV₃Sb₃. To reach these conclusion (and to distinguish the CsV₃Sb₅ system from the phase coherence scenario in previous PdCoO₂ family), the authors have stressed on the "quantum mean free path" derived from the Dingle temperature, which, however, is entirely different from the true phase coherence length as shown below:

The transport mean free length L_m , the quantum scattering length (L_q , derived from Dingle temperature, or quantum mean free path denoted by the authors in the rebuttal) and the phase coherence length (L_ϕ) depict entirely different scattering processes:

1. L_m is the average distance an electron travels between any scattering event (either elastic or inelastic), and is somehow insensitive to small-angle scattering (only large-angle scattering contributes strongly). This value is weakly temperature-dependent.

2. L_q is the average distance an electron travels before its quantum phase is randomized by elastic scattering. Corresponding scattering time accounts for all elastic scattering events regardless of its scattering angle. Note this is also the reason why L_q is consistently much smaller than L_m in most reported systems including the CsV₃Sb₅ here, as small-angle scattering is usually common in materials.

3. L_ϕ is the maximum distance an electron retains quantum phase coherence, which is governed by inelastic (e.g., e-e or e-p) scattering. This value is strongly temperature-dependent. At low temperatures, L_ϕ can be significantly larger than L_q , L_m or even the sample size, as in the case of 2D electron gas.

The obfuscation of L_q and L_ϕ , and the resultant wrong estimation of true phase coherence length in the article severely limit the scientific soundness, making the critical declaration on page 7 ("Thus, the phase coherence length may well exceed the transport m.f.p., yet this does not hold for CsV₃Sb₅") hold no water.

We thank the reviewer for the discussion of scattering length scales. Indeed, we have not discussed the dephasing length L_ϕ . It is exciting to see such strong support from the referee for the scenario of $L_\phi \gg W$. We had omitted its discussion as it implies a micron-sized, 2D phase-coherent quantum diffusion at 30K, a phenomenon usually studied in the sub-Kelvin range. This in turn implies the astonishing absence of inelastic scattering despite substantial broadening of the Fermi-Dirac distribution or electron-phonon coupling (which is known to be finite from the fact this is a charge-density-wave material). This forms the rationale for our

discussion limiting to classical or quantum ballistic behavior, yet we fully agree that a quantum diffusive picture could be mentioned.

"A further scenario based on coherent quantum diffusion in the presence of elastic scattering, in principle, may also reconcile the observation of interference at length scales above a transport mean free path. This scenario at elevated temperatures implies the absence of inelastic scattering despite the substantial phonon density and Fermi-Dirac broadening, which is highly unlikely given the strong electron-phonon coupling in the material (M. Gutierrez-Amigo et al., *Communications Materials* 5, 234 (2024)/M. Alkorta et al., arXiv:2505.19686 (2025))."

Moreover, a microscopic model remains elusive, and the proposed many-body related physics have been frequently reported by previous STM and many other probes (Refs.46,20,21,22, etc.).

We agree that the many-body physics in CsV₃Sb₅ has been proposed previously. Yet, the main scope of our manuscript is to demonstrate the experimental observation of **many-body electron coherence**, where the quantum coherence exceeds the single-particle picture and surprisingly survives despite the presence of electron-electron interactions. This is unambiguously demonstrated with our observations of phase-coherent transport in CsV₃Sb₅. We believe this offers a vital ingredient that is currently missing for reconciling the microscopic origin of the cascades of correlated electronic orders in Kagome metals.

As a result, I do not see sufficient soundness and depth of the study to justify the publication in Nature. Nevertheless, the experimental part of this work is interesting and certainly suited for a high-impact journal on lower general importance level than Nature after necessary changes.

We thank the reviewer for their positive assessment of our work for a high-impact journal. The main critique is the omission of the possibility of coherent phase diffusion at 30K, which is switchable by field and follows the geometric shape of the sample – an entirely novel state of quantum matter at odds with our current understanding of metals.

Referee #2 (Remarks to the Author):

I co-reviewed this manuscript with one of the reviewers who provided the listed reports.

Referee #3 (Remarks to the Author):

The updated manuscript by Guo et al. addresses many of the issues raised by reviewers in the first round. Of particular importance, this draft highlights the key differences between the observation of periodic in magnetic field magnetoresistance oscillations in CsV₃Sb₅ and (Pd,Pt)CoO₂. While it is still mechanistically unknown what is driving these oscillations in CsV₃Sb₅, with the new data presented here on parallelogram-shaped bars it is clear that this result cannot be understood by a simple independent single-particle state picture. As such, we recommend the publication of this work in Nature.

We thank the referee for their comments. We are pleased that the new results further demonstrate the presence of quantum coherence beyond the single-particle regime in CsV₃Sb₅, which is the main point of the manuscript.

One important typo that needs to be fixed: at the bottom of page 5 a part of a sentence seems to be missing: “A striking result emerged: while the large oscillations again follow the same $\cos(\phi) - 1$ scaling, set by the different widths of both sidewalls(Fig. 2d).”

Thank you for pointing this out. The typo has now been clarified.

Referee #4 (Remarks to the Author):

I co-reviewed this manuscript with one of the reviewers who provided the listed reports.

Referee #5 (Remarks to the Author):

The authors have provided thorough answers to my questions in the last review, where they have included much more detailed discussions and analyses. I recommend the publication of this manuscript on Nature.

We are glad that the referee is satisfied with our response to their comments; indeed, they are very helpful for the improvement of the revised manuscript. And we thank the referee for their recommendation.